# Effects of Community-Based Programs on Integration into the Mental Health and Non-Mental Health Communities

**DOI:** 10.3390/healthcare9091181

**Published:** 2021-09-08

**Authors:** Mi Kyung Seo, Min Hwa Lee

**Affiliations:** 1Department of Social Welfare, Gyeongsang National University, Jinju 52828, Korea; welseo@gnu.ac.kr; 2Department of Social Welfare, Mokpo National University, Muan 58554, Korea

**Keywords:** mental health community integration, non-mental health community integration, resource accessibility, mental health service programs

## Abstract

**Aims**: The purpose of this study was to verify how integration into the mental health community, a subculture of persons with mental illness, affects the integration into the non-mental health community. Thus, we analyzed the effect of community-based mental health service programs on non-mental health community integration, mediated by mental health community integration. **Methods:** In total, 190 persons with mental illness (M age = 42.78; SD = 11.3; male, 54.7%; female, 45.3%), living in local communities and using community-based mental health programs, participated in the study. We measured their sociodemographic and clinical variables, the environmental variables of mental health service programs, and the level of integration of the mental health and non-mental health communities. The data collected were analyzed to test the proposed hypotheses using Structural Equation Modeling (SEM). **Results:** The common significant predictors affecting the two types of community integration were symptoms and resource accessibility: the more accessible the various community resources and the less severe the psychiatric symptoms were, the higher the level of the two types of community integration was. In path analysis, the program’s atmosphere and the participation of people with mental illness (program involvement) significantly predicted the level of integration into the mental health community. This, in turn, had a positive effect on their physical integration, social contact frequency, and psychological integration into the non-mental health community, mediated by the integration of the mental health community. **Conclusion:** Based on the results, we emphasize the importance of mental health communities and suggest strategies to support the integration of mental health communities.

## 1. Introduction

Deinstitutionalization has led mental health treatment facilities to shift from facilities within hospitals to within communities. As a result, the purpose of mental health services is to enable people with mental disorders to satisfactorily participate in their communities with equal rights [1]. For proper recovery, above all, full integration into the community, which is the context and field of their lives, must be the goal. Community integration is, therefore, both a facilitator and an outcome of the recovery process [2,3]. In their definition of community integration, Wong and Solomon [4] included the individual’s capacity to carry out daily activities in their community (physical integration), to pursue interaction with mentally well members of their community (social integration), and to feel a sense of belonging within their community (psychological integration). Therefore, a normalization strategy should be pursued that allows people with mental illness to actively interact with non-disabled people, participate in various community activities with equal rights, and build psychological bonds within their community.

Many communities, however, are not friendly in their interactions with those suffering from mental illness who wish to integrate. Social networks for people with mental illness are small and provide little social support. Due to social stigma, they have limited opportunities for employment, housing, and education [3,5,6]. Only 20% of people with mental illness are employed in full-time jobs, and 60% say they do not even try to obtain a job for fear of unfair treatment [7]. In Korea, their social exclusion is even more severe. The employment rate for the entire population aged 15 or older is 60.2%, while the rate for persons with mental disorders is 9.9% [8]. Moreover, 10.2% of people with mental disorders suffer from housing insecurity and only 2.4% use community services [9]. Even when they attempt to utilize medical services, people with mental illness have higher mortality rates than the general population due to discrimination [10,11]. Unfortunately, people with mental illnesses are still socially excluded, which contradicts the purpose of mental health services.

Today, the concept of community has extended beyond locality to a non-place concept [12]. Some communities share hobbies and interests, and can sometimes be formed on the basis of identities, including race, culture, disorders, etc. Chinatown is a representative example. Communities formed within subcultures not only have common interests but also have similar experiences within the larger dominant society. Thus, people interact more often within a subcultural community whose members share the characteristics with which they identify [13].

Persons with mental illness also belong to various communities based on their diverse identities (family, workplace, religion, club, neighborhood, etc.). Wong et al. [12] conducted a focus group study to examine the perspectives of mental health consumers on the concept of community. The following four types of communities emerged: cultural identity, faith community, treatment community, and neighborhood. The following two core domains are common to the four types of communities: (1) the togetherness of contributing to the community as members and (2) community acceptance without being rejected, regardless of mental health status. Of the four communities they identified, the treatment community can be understood as the activities of a mental health service program. In addition to the core domains, this is a community that includes wellness management to deal with needs related to the participants’ physical, emotional, and behavioral health problems, and various activities related to psychosocial rehabilitation. For people with mental illness, therefore, the treatment community is where they can feel a sense of belonging and be accepted without being rejected, and it includes various activities that address their needs. 

Mental health communities comprising mental health service programs are vital sources of psychological and social support. In most research, the goal of community integration is to fully integrate persons with mental illness into broader communities. Integration within the mental health community, in which people with mental illness interact closely with each other, has been regarded as socially isolating [12] and as an indication that the individuals are still ill [13].

The goal of mental health services is normalization, which is possible only through integration into broader, non-mental health communities. One way of viewing community integration is to see it as assimilation into the entire society by melding all the characteristics. In contrast to this viewpoint, there is another interpretation of community integration, namely as a patchwork quilt or mosaic, in which people can participate in a larger society, while retaining their own cultural identities [14]. According to this viewpoint, integrating persons with mental illness into the mental health community can provide them with opportunities to play healthy roles in an accepting environment and facilitate their integration into broader society [15]. Togetherness and psychological support within a subculture can boost self-esteem [13], and collective identity can be a counterforce to discrimination [16]. It is, therefore, necessary to verify whether integration into a mental health community will facilitate or hinder integration into the general community.

Previous studies on community integration have focused on integration into non-mental health communities, in line with the normalization ideology, and on identifying the factors that predict it. In general, sociodemographic and clinical variables [2,17,18], social context variables [19,20], community-based mental health program variables [4,21,22,23], and community variables [24,25,26] such as community acceptance, stigma, and the neighborhood environment were found to be significant predictors of the integration of people with mental illness into general communities. These studies, however, overlooked the importance of the integration of people with mental illness into mental health communities based on their subculture. Therefore, this study analyzed the effects of the mental health service program environment on the non-mental health community integration of people with mental illness, mediated by their mental health community integration. 

## 2. Method

### 2.1. Participants

The Institutional Review Board of Gyeongsang National University approved this study. We recruited 190 participants with mental disorders over the age of 20 years. The inclusion criteria for participants included a diagnosis of schizophrenia or a mood disorder according to DSM-5, living within the community, and using community-based mental health service programs. We administered a survey to individuals who understood the purpose of the study and who voluntarily submitted written informed consent to participate.

The sociodemographic characteristics of the 190 participants are shown in Table 1. Among the participants, 104 (54.7%) were male and 86 (45.3%) were female. The average age was 42.78 (±11.3); 33 participants were 20–29 years old (17.4%), 40 were 30–39 years old (21.1%), 54 were 40–49 years old (28.4%), 53 were 50–59 years old (27.9%), and 10 participants were over 60 years old (5.3%). The average number of years of education completed was 15.76 (±2.88 years). In total, 109 (57.4%) participants were high school graduates and 51 (26.8%) of the participants were employed: 12 (6.3%) were full-time workers, 26 (13.7%) were part-time workers, and 13 (5.3%) were vocational rehabilitation workers. Participants with schizophrenia made up 157 (82.6%) of all participants, followed by 15 with bipolar disorder (7.9%), and 13 with major depression (6.8%). Of the 190 participants, 130 (68.4%) lived in large cities and 60 (31.6%) lived in small cities.

### 2.2. Measures

#### 2.2.1. Integration into the Mental Health Community

To measure mental health community integration (MHC), we used a modified version of the Community Integration Measure (CIM) that was altered to fit the Korean context. The CIM is a scale developed by McColl et al. [27] to measure subjective experiences of community integration from the perspective of service consumers. It consists of 10 items divided into the following two factors: belonging and independent participation. For this study, we changed “community” in the original scale to “mental health facilities”. Each item was rated from 1 (completely not true) to 5 (completely true), where a higher score indicated higher levels of integration into the respondents’ mental health community. The Cronbach’s α for mental health community integration was 0.908.

#### 2.2.2. Integration into the Non-Mental Health Community

##### Physical Integration

To measure physical integration (PHI), we used the External Integration Scale [28], which was modified by Aubry and Myner [29] and Choi [30]. Thirteen items assessed the individual’s frequency of involvement in different outside activities, such as eating in a restaurant, visiting a library, walking in a park, sending messages, using social media, and calling to say hi. Each item was rated from 1 (never) to 5 (very often), in which higher scores suggested higher levels of physical integration. The Cronbach’s α for physical integration was 0.821.

##### Social Integration

Social integration assessed the number of social relationships by social network size and social contact frequency. Social network size (SN) was measured by the number of families, relatives, friends, neighbors, or peers they had been in touch with over the past year. Items in the social contact frequency (CF) asked respondents how often they had been in different types of social contact with families, relatives, friends, neighbors, or peers over the past year, ranging from relatively superficial (such as saying hello) to closer contact (such as going out together). Social contact frequency was scored from 1 (never) to 9 (almost every day), with higher scores indicating higher levels of social integration. The Cronbach’s α for social integration by social network size was 0.916 and that for social contact frequency was 0.547.

##### Psychological Integration

In order to measure psychological integration (PSI), we used the Neighborhood Cohesion [31]. Ten items, assessing the perceived sense of community belonging, were rated from 1 (completely not true) to 5 (completely true), in which higher scores suggested higher levels of psychological integration. The Cronbach’s α for psychological integration was 0.834.

#### 2.2.3. Community-Based Program Environment

##### Program Atmosphere

To measure the program’s atmosphere (PA), we used Moos’s Community-Oriented Program Environment Scale (COPES) [32] standardized by Booth [33]. Nine items assessed the structure, organization, and clarity of the participants’ roles and the expectations for them within their programs. Each item was rated from 1 (completely not true) to 5 (completely true), in which higher scores suggested a more positive program atmosphere; in other words, the program is systematically structured so that it can be clearly understood, and it is recognized by participants that involvement in the program has a positive effect. The Cronbach’s α for program atmosphere was 0.893.

##### Relationships with Patients and Staff

We also used the COPES standardized by Booth [33] to measure the participants’ relationships with other patients and staff members. Relationships with patients (RP) consisted of seven items measuring sharing concerns, intimacy, emotional expression, information sharing, and a sense of belonging. Relationships with staff (RS) also consisted of seven items measuring encouragement and advice for members and attitudes toward questions and answers. Each item of the two relationship scales was rated from 1 (completely not true) to 5 (completely true), where higher scores indicated more positive relationships with patients and staff. The Cronbach’s α for relationships with patients was 0.745 and that for relationships with staff was 0.784.

##### Program Involvement

To evaluate program involvement (PI), we used the program participation scale, which was modified by Lee [34]. This scale consists of nine items, measuring expectations of involvement in the program, efforts towards the program, and program satisfaction. Each item was rated from 1 (completely not true) to 5 (completely true), where higher program involvement scores indicated more active program involvement and greater program satisfaction. The Cronbach’s α for program involvement was 0.770.

#### 2.2.4. Sociodemographic and Clinical Variables

We assessed all participants’ sociodemographic variables, including gender, age, years of education, average monthly income, city size, resource accessibility, and clinical variables, including diagnoses and psychiatric symptoms. To measure resource accessibility, we used the Access Scale of Kim [35], which was modified from Segal and Aviram’s External Integration Scale [28]. It consists of nine items examining how easily respondents can use parks, libraries, movie theaters, etc. Each item was rated from 1 (completely not true) to 5 (completely true), in which higher scores indicated more diverse and accessible community resources. The Cronbach’s α for resource accessibility was 0.885.

Symptoms were evaluated based on the degree to which patients had recently experienced psychiatric symptoms. To measure symptoms, we used the Colorado Symptom Index, which was adapted to the Korean context by Lee and Seo [36]. This scale consists of 14 items that evaluate the extent to which patients experience hallucinations, delusions, memory loss, suicidal thoughts, and mood disorders. Each item was measured on a 5-point scale ranging from 1 (not at all) to 5 (strongly agree); the higher the score is, the higher the level of symptoms is. The Cronbach’s α for symptoms was 0.889.

### 2.3. Statistical Analysis

Statistical analyses were performed using SPSS 26.0 and AMOS 26.0 (SPSS Inc., Chicago, IL, USA). To review the basic assumptions of regression before the analysis, we examined outliers, normality, and multi-collinearity. To verify the reliability of the scale, Cronbach’s alpha was used. Descriptive statistical analysis was conducted to examine the sociodemographic characteristics of the participants. Regression analysis was used to examine the effects of the sociodemographic and clinical variables on the two types of community integration.

Structural equation modeling (SEM), which involves a measurement model and a structural model, was used to analyze the cause–effect relationships among latent variables. SEM was used to analyze the effects of the mental health service program environment on general community integration, mediated by mental health community integration. Using a two-step approach, confirmatory factor analysis (CFA) was first conducted to test the validity of the measurement model, which examined the relationships between the observed and latent variables. The next step was to test the structural model and improve the goodness of fit by using modification indices. The chi-square test, absolute fit measures, and incremental fit measures were considered together to validate the model’s goodness of fit. After the optimized model had been derived and then confirmed, the significant influencing factors and the regression weights were evaluated.

## 3. Results

### 3.1. Effects of Sociodemographic and Clinical Variables on Integration into the Mental Health and Non-Mental Health Communities

The effects of the sociodemographic and clinical variables on the two types of community integration were analyzed using regression analysis (Table 2). For mental health community integration, the model’s explanatory power was 39.2% and the model’s fit was statistically significant (F = 16.594, *p* = 0.000). Resource accessibility (β = 0.537) and symptoms (β = −0.191) were significant predictors of mental health community integration. For those with higher resource accessibility and fewer symptoms, the level of mental health community integration was higher. 

For non-mental health community integration, concerning physical integration, the explanatory power of the model was 35.2% and the model’s fit was statistically significant (F = 13.948, *p* = 0.000). The level of physical integration was higher for greater resource accessibility (β = 0.468), living in a large city (β = 0.188), and experiencing fewer symptoms (β = −0.136). For social integration, the explanatory power of the model was 19.9% for social network size and 11.2% for social contact frequency, and the models’ fit was statistically significant (F = 6.369, *p* = 0.000; F = 3.257, *p* = 0.003). The first type of social integration, social network size, was significantly affected by years of education (β = 0.212), resource accessibility (β = 0.238), and symptoms (β = −0.232). The second type of social integration, social contact frequency, was significantly affected only by resource accessibility. With greater resource accessibility (β = 0.241), social contact frequency increased. Finally, regarding psychological integration, the model’s explanatory power was 25.5% and the model’s fit was statistically significant (F = 8.783, *p* = 0.000). Resource accessibility (β = 0.331) and symptoms (β = −0.264) also predicted psychological integration, with higher psychological integration levels occurring in the participants who had greater resource accessibility and fewer psychiatric symptoms.

### 3.2. Confirmatory Factor Analysis 

Confirmatory factor analysis (CFA), which tests a measurement model by focusing on the relationship between the observed variables of a particular latent variable, was conducted to assess the observed variables with factor loadings less than 0.5 and to confirm the fit of the measurement model. Through the CFA, the observed variables (ct4, pro4, p4, ph4, so3, sf1, sf2) with factor loadings less than 0.5 were deleted. All the factor loadings of the observed variables were over 0.5, as shown in Figure 1. There were 26 observed variables in the measurement model. The goodness-of-fit indices of the full measurement model are presented in Table 3. According to Table 3, the chi-square test was found to be inappropriate, but the other indices might be considered together. Most of the indices met their corresponding acceptable requirements. 

### 3.3. Research Model Verification

Following the CFA, the structural model was composed of nine constructs with several observed variables. The initial research model’s results did not fit the data very well. To modify the initial research model, correlations were made between the structural errors of the latent variables, as the modification indices suggested. The final research model shown in Figure 2 was derived. The goodness of fit of the final research model was evaluated to verify the effects of the mental health service program environment on general community integration, mediated by mental health community integration, as shown in Table 4. According to Table 4, the chi-square test was found to be inappropriate, but the other indices might be considered together. Most of the indices met their corresponding acceptable requirements. 

From Table 5, it can be seen that five paths are significant at the 0.05 or 0.001 level: the path from PA to MHC, the path from PI to MHC, and the paths from MHC to PHI, CF, and PSI. Therefore, program atmosphere and program involvement significantly affected physical integration, social contact frequency, and psychological integration into the mental health community. The path regression weight from PA to MHC was 0.410, that from PI to MHC was 0.439, that from MHC to PHI was 0.441, that from MHC to CF was 0.260, and that from MHC to PSI was 0.542. In other words, the members who participated more frequently and diligently in positive programs had higher levels of mental health community integration. This had a positive effect on diverse community activities (physical integration), social contact frequency with others (social integration), and a sense of belonging within the community (psychological integration).

The result of examining the relative influence of each variable through standardized regression weights was that integration into mental health communities had the greatest impact on psychological integration (β = 0.542). This was followed by the effects of integration into the mental health community on physical integration (β = 0.441), the effects of program involvement on integration into the mental health community (β = 0.439), the effects of program atmosphere on integration into the mental health community (β = 0.410), and the effects of integration into the mental health community on social contact frequency (β = 0.260). 

## 4. Discussion

The purpose of this study was to verify whether mental health community integration, based on the subculture of people with mental illness, would help or hinder their integration into the wider community. To achieve this aim, we analyzed the effects of the respondents’ characteristics (sociodemographic and clinical) on the two types of community integration and conducted path analyses to investigate the effects of a mental health program’s environment on integration into non-mental health communities, mediated by their mental health community integration. Several important findings were identified. 

First, we analyzed the influence of sociodemographic and clinical variables on the four areas of integration into the non-mental health community (physical, social network size, social contact frequency, and psychological integration) and into the mental health community, which is a subculture of people with mental illness. It emerged that the predictors for all the integration areas were similar. The most common predictors of both types of community integration analyzed were high resource accessibility among the demographic variables and low symptom levels among the clinical variables. Gender, age, and years of education had no significant predictive power. In short, greater resource accessibility and fewer psychiatric symptoms led to higher levels of integration into the two communities. 

Some inconsistent results appeared regarding the relevance of gender or age to community integration. Some results have shown that gender and age are not related to integration [18,21,37], but others have shown that women have higher levels of integration than men, and that younger people have higher levels than older people [17,38]. However, symptoms and resource accessibility were significant predictors in most studies [2,4,39,40]. Therefore, to improve the integration of people with mental illness into both communities, it is necessary to reduce their symptoms and to ensure the availability and accessibility of community resources. What we considered unusual was that physical integration, a subdomain of non-mental health community integration, was predicted by community size. The larger the community is, the greater the physical integration into the non-mental health community is. This finding was different from that of Kruzich [17], who found that small cities had more integration. However, since the characteristics of cities vary from country to country, additional studies reflecting cultural characteristics are required.

Second, we analyzed the effects of community-based service programs’ environments on four subdomains of non-mental health community integration, mediated by mental health community integration. We identified a significant path by which program atmosphere and involvement among program environmental factors affected physical integration, social contact frequency, and psychological integration, mediated by mental health community integration. Previous researchers [21,22,23] have noted the effects of a facility’s environment and program variables on non-mental health community integration but have reported that their effects are not great. Pahwa et al. [41] analyzed the relationship between the intensity of the services provided and the two types of community integrations. They found that mental health community integration was higher in the group that received high-intensity services, but non-mental health community integration was higher in the group that received low-intensity services. Based on these results, the researchers considered that the people who received high-intensity services were less normalized. Unlike their study, we focused on the significant influence that the program environment had on general community integration, mediated by integration into the mental health community. These results support the argument that integration into mental health service facilities provides patients with opportunities to gain social support, regain confidence, and integrate into the wider dominant society [12,13,14].

Our results that mental health community integration significantly predicted non-mental health community integration have several implications. The first implication is that the sense of belonging and togetherness within the mental health community, which is a subculture, does not hinder integration into mainstream society, but instead can promote it. This demonstrates that interactions within subcultures can contribute to the recovery of people with mental illness rather than isolating them from the public.

The second implication is that the content of the program is more important than the relationships within the program, because the program atmosphere and program involvement were the most significant predictors of mental health community integration among the program environment variables. That is, programs that are systematically structured, with clearly specified roles for and expectations of the participants, and programs that participants actively engage in improve the participants’ integration into the mental health community. Lee and Seo [42] also identified that program atmosphere was the most significant predictor of life satisfaction and social adaptation among people with mental disorders.

Third, mental health community integration had the least effect on social integration among the subdomains of non-mental health community integration. Not only did the social network size have no significant influence, but it also had the least influence on social contact frequency. Social integration is a particularly vulnerable area for individuals with mental illness. Lee and Seo [6] compared community integration levels between the general population and people with mental illness and found significantly lower social integration levels (social network size and social contact frequency) among those with mental illness. As social integration is one of the factors that best predict the quality of life [37,43], efforts must be made to improve this.

## 5. Conclusions

Based on these results, we suggest the following. First, the two factors that commonly predicted both types of community integration were psychiatric symptoms and resource accessibility. Resource accessibility means the variety of community resources and how easily they are available. If accessibility is high, most of the integration levels will increase. However, in Korea, most mental health services and resources are concentrated in large cities, and thus, there is a wide gap in resource accessibility between regions. This problem can be mitigated through community development policies that equalize resources by distributing them evenly across regions. Before that reality takes place, it may be possible to develop resources directly within the mental health service system. For instance, operating a restaurant or cafe directly in the community as supportive employment, vocational rehabilitation, or a social enterprise could be a strategy to increase resource accessibility and the participation of people with mental illness at the same time. These business incubators have been proposed as alternatives to community development [14]. Symptoms are as important a predictor as resource accessibility: The lower the symptom levels are, the higher the integration levels are in most cases. We cannot overemphasize the importance of psychiatric symptom management for achieving any mental health service’s goals, including community integration. It may also have a positive impact on reducing the social cost of mental illness. 

Second, because mental health community integration can positively predict non-mental health community integration, it is necessary to systematically structure the programs and encourage participants to participate actively. It is important not to attempt to integrate participants hastily into their broader communities in an attempt to enforce the normalization ideology. Their experience and confidence of playing a role without discrimination within the mental health community, which is their subculture, may empower those with mental disorders to resist unfair treatment by the dominant society and actively explore their social roles. Of course, if the subject wishes, they may stay within their mental health community without attempting broader community integration, and it is important to respect this choice.

## 6. Limitations

Despite the value of our findings, we must note the certain limitations of this study. First, mental health community integration can be understood as an intermediate step in the course of non-mental health community integration; in other words, the former precedes and gradually evolves into the latter. To understand this process, longitudinal studies are needed to track the subjects’ changes over time, but we could not analyze the temporal changes by evaluating integration into the two communities at the same time. Second, in this study, the evaluation of program environment relied only on the participants’ subjective experiences, lacking access to objective information such as actual program numbers and content, participant numbers, staff-to-client ratios, etc. When evaluating a program, it is necessary to consider objective data as much as the subjective evaluations of the participants.

## Figures and Tables

**Figure 1 healthcare-09-01181-f001:**
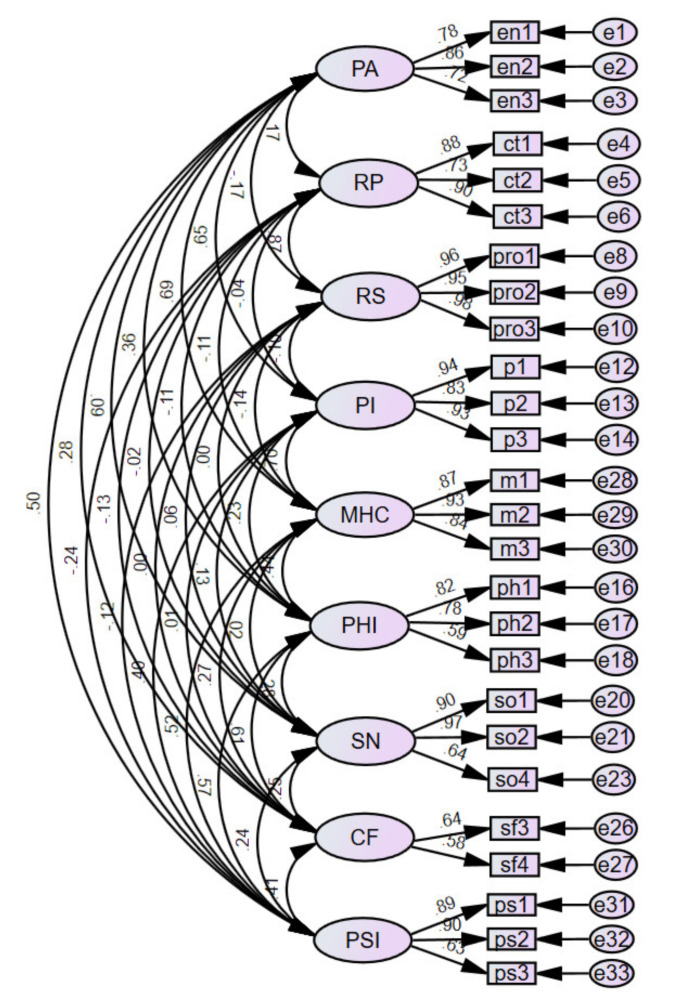
Standardized regression weights of the full measurement model. PA: program atmosphere; RP: relationships with patients; RS: relationships with staff; PI: program involvement; MHC: integration into the mental health community; PHI: physical integration; SN: social network size; CF: social contact frequency; PSI: psychological integration.

**Figure 2 healthcare-09-01181-f002:**
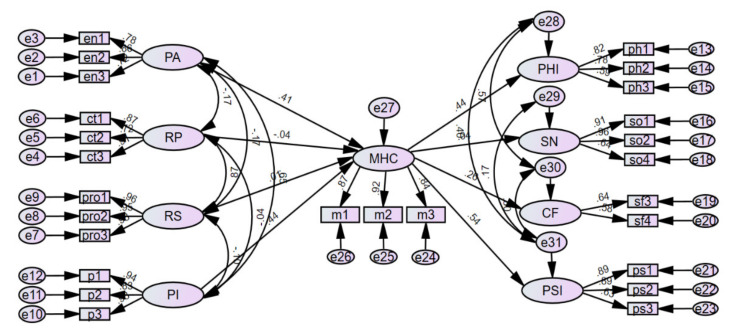
Standardized regression weights of the final research model. PA: program atmosphere; RP: relationships with patients; RS: relationships with staff; PI: program involvement; MHC: integration into the mental health community; PHI: physical integration; SN: social network size; CF: social contact frequency; PSI: psychological integration.

**Table 1 healthcare-09-01181-t001:** Sociodemographic characteristics of the participants.

Variable	Category	Frequency	Percentage (%)
Gender	Male	104	54.7
	Female	86	45.3
Age	20–29	33	17.4
	30–39	40	21.1
	40–49	54	28.4
	50–59	53	27.9
	Over 60	10	5.3
Education level	Middle school or below	19	10.0
	High school	109	57.4
	University or over	60	31.6
	Other	2	1.0
Diagnosis	Schizophrenia	157	82.6
	Major depression	15	7.9
	Bipolar disorder	13	6.8
	Other	5	2.6
Employment status	Vocational rehabilitation	13	6.8
	Part time	26	13.7
	Full time	12	6.3
	No job	139	73.2
City size	Large cities	130	68.4
	Small cities	60	31.6

**Table 2 healthcare-09-01181-t002:** Effects of sociodemographic and clinical variables on integration into the mental health and non-mental health communities.

Variables		Mental Health Community Integration (1)			Non-Mental Health Community Integration											
					Physical Integration (2)			Social Integration						Psychological Integration (5)		
								Social Network Size (3)			Social Contact Frequency (4)					
		β	t	*p*	β	t	*p*	β	t	*p*	β	t	*p*	β	t	*p*
**Sociodemographic**																
	Gender	−0.030	−0.512	0.609	−0.101	−1.656	0.099	−0.032	−0.470	0.639	−0.055	−0.768	0.443	−0.095	−1.446	0.150
	Age	0.054	0.884	0.378	0.020	0.324	0.746	0.029	0.413	0.680	−0.025	−0.332	0.740	0.084	1.245	0.215
	Years of education	−0.034	−0.558	0.577	0.072	1.158	0.248	0.212	3.036	0.003	0.135	1.840	0.067	0.068	1.019	0.310
	City size	0.047	0.798	0.426	0.188	3.088	0.002	0.006	0.089	0.929	0.037	0.513	0.609	0.092	1.410	0.160
	Resource accessibility	0.537	8.479	<0.000	0.468	7.160	<0.000	0.238	3.277	0.001	0.241	3.150	0.002	0.331	4.722	<0.000
**Clinical**																
	Diagnosis	−0.059	−0.956	0.340	0.039	0.615	0.539	0.038	0.541	0.589	0.032	0.433	0.666	0.058	0.858	0.392
	Symptoms	−0.191	−3.154	0.002	−0.136	−2.181	0.031	−0.232	−3.332	0.001	−0.084	−1.145	0.254	−0.264	−3.934	<0.000

Gender: 1 = male; city size: 1 = large city; diagnosis: 1 = schizophrenia. (1) F = 16.594, df = 7; 180, R^2^ = 0.392, *p* < 0.001. (2) F = 13.948, df = 7; 180, R^2^ = 0.352, *p* < 0.001. (3) F = 6.369, df = 7; 179, R^2^ = 0.199, *p* < 0.001. (4) F = 3.257, df = 7; 180, R^2^ = 0.112, *p* < 0.01. (5) F = 8.783, df = 7; 180, R^2^ = 0.255, *p* < 0.001.

**Table 3 healthcare-09-01181-t003:** Goodness of fit of the measurement model.

$x^{2}=425.809\;(p<0.001);\;\mathrm{Degrees}\;\mathrm{of}\;\mathrm{Freedom}=263\;(351-88)$			
Goodness-of-Fit Measure		Level of Acceptable Fit	Fit Statistics
Absolute fit	$x^{2}$/df	<3 good	1.619
	GFI	>0.8 acceptable, >0.9 good	0.858
	AGFI	>0.8 acceptable, >0.9 good	0.810
	RMSEA	<0.08 good	0.057
Incremental fit	NFI	>0.9 good	0.894
	RFI	>0.9 good	0.869
	IFI	>0.9 good	0.957
	TLI	>0.9 good	0.946
	CFI	>0.9 good	0.956

**Table 4 healthcare-09-01181-t004:** Goodness of fit of the final research model.

$x^{2}=478.486\;(p<0.001);\;\mathrm{Degrees}\;\mathrm{of}\;\mathrm{Freedom}=281\;(351-70)$			
Goodness-of-Fit Measure		Level of Acceptable Fit	Fit Statistics
Absolute fit	$x^{2}$/df	<3 good	1.703
	GFI	>0.8 acceptable, >0.9 good	0.842
	AGFI	>0.8 acceptable, >0.9 good	0.803
	RMSEA	<0.08 good	0.061
Incremental fit	NFI	>0.9 good	0.881
	RFI	>0.9 good	0.863
	IFI	>0.9 good	0.947
	TLI	>0.9 good	0.938
	CFI	>0.9 good	0.947

**Table 5 healthcare-09-01181-t005:** Regression weights in the final research model.

			B	β	S.E.	C.R.	*p*
PA	→	MHC	0.460	0.410	0.102	4.517	<0.001
RP	→	MHC	−0.021	−0.041	0.069	−0.300	0.764
RS	→	MHC	0.004	0.009	0.050	0.071	0.943
PI	→	MHC	0.278	0.439	0.053	5.285	<0.001
MHC	→	PHI	0.624	0.441	0.118	5.282	<0.001
MHC	→	SN	0.508	0.045	0.884	0.575	0.566
MHC	→	CF	0.843	0.260	0.340	2.483	0.013
MHC	→	PSI	0.637	0.542	0.090	7.087	<0.001

B: regression weights; β: standardized regression weights; PA: program atmosphere; RP: relationships with patients; RS: relationships with staff; PI: program involvement; MHC: integration into the mental health community; PHI: physical integration; SN: social network size; CF: social contact frequency; PSI: psychological integration.

## Data Availability

Restrictions apply to the availability of these data. Data was obtained from the participants and are available with the permission of the participants.

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
