# Peer review of "Effects of Community-Based Programs on Integration into the Mental Health and Non-Mental Health Communities"

_healthcare, 2021, doi:10.3390/healthcare9091181_

Round 1
Reviewer 1 Report
- It is somehow confusing to use the wording “mental health community” for persons with mental illness, while “non-mental health community” seems used for the cognitively healthy group? Why not the other way round?
- Analyses should deal with the typically large interindividual differences of people with cognitive impairments. Besides that, there is a broad age range that further adds variance. Thus, are the observed patterns moderated by age, education, economics, physical health issues, etc.? (also including e.g. moderated mediation analysis for the path models)
- The practical implications could be discussed in more detail. What about the cost-effectiveness ratio?
Reviewer 2 Report
P1, L45: missing word between for and with
P4: Not sure these are outdoor activities, perhaps “outside” for outside the program. Subtle difference in English.
P4, L158: Neighborhood Cohesion [Word Missing]
Author Response
We would like to express our appreciation to the reviewer who provided some constructive suggestions to improve our manuscript.
p.1 (L45) : enter a word "people"
p.4 (L142) : change "outdoor" to "outside"
p.4 (L158) : "Neighborhood Cohesion"
Notation written in the original article
Reviewer 3 Report
It is a good manuscript. I sugguest to accept it after making minor revisions.
This study aimed at verifying how integration into the mental health community, a subculture of persons with mental illness, affects the integration into the non-mental health community. Thus, we analyzed the effect of community-based mental health service programs on non-mental health community integration mediated by mental health community integration. The acceptance of manuscript would depend on making minor revision:
- The question and the aim of this study are original and they are well defined.
- The results and the discussion are well defined.
- Limitation and policy implications should be separated from conclusions.
- The quality of presentation had met many standards of presenting.
- The manuscript has an interest for the readers. It is an evaluation of community care integration for mental health patients and it might under minor revisions to have an international response.
- Community care integration is a core issue in mental health. So, a manuscript describing the evaluation of mental health programmes in primary health care might have an overall merit. It increases the knowledge in community care for mental health patients.
- English language is not appropriate and understandable. Special editing is necessary.
Author Response
We would like to express our appreciation to the reviewer who provided some constructive suggestions to improve our manuscript.
p.11 : Limitation has separated from conclusions.
And we have used English editing from MDPI.